# A Herpesvirus of Turkey-Based Vector Vaccine Reduces Transmission of Newcastle Disease Virus in Commercial Broiler Chickens with Maternally Derived Antibodies

**DOI:** 10.3390/vaccines8040614

**Published:** 2020-10-16

**Authors:** Timea Tatár-Kis, Egil A.J. Fischer, Christophe Cazaban, Edit Walkó-Kovács, Zalan G. Homonnay, Francisca C. Velkers, Vilmos Palya, J. Arjan Stegeman

**Affiliations:** 1Scientific Support and Investigation Unit, Ceva-Phylaxia, 1107 Budapest, Hungary; timea.tatar@ceva.com (T.T.-K.); edit.kovacs@ceva.com (E.W.-K.); zalan.homonnay@ceva.com (Z.G.H.); vilmos.palya@ceva.com (V.P.); 2Population Health Department, Veterinary Medicine Faculty, Utrecht University, 3584 CL Utrecht, The Netherlands; e.a.j.fischer@uu.nl (E.A.J.F.); F.C.Velkers@uu.nl (F.C.V.); 3Science and Investigation Department, Ceva Animal Health, 33500 Libourne, France; christophe.cazaban@ceva.com

**Keywords:** recombinant vaccine, Newcastle disease, transmission, challenge experiment, shedding control, poultry, reproduction number, R

## Abstract

Newcastle Disease is one of the most important infectious poultry diseases worldwide and is associated with high morbidity, mortality, and economic loss. In several countries, vaccination is applied to prevent and control outbreaks; however, information on the ability of vaccines to reduce transmission of ND virus (NDV) is sparse. Here we quantified the transmission of velogenic NDV among 42-day-old broilers. Chickens were either vaccinated with a single dose of a vector vaccine expressing the F protein (rHVT-ND) at day-old in the presence of maternally derived antibodies or kept unvaccinated. Seeders were challenged 8 h before the co-mingling with the corresponding contacts from the same group. Infection was monitored by daily testing of cloacal and oro-nasal swabs with reverse transcription-real-time PCR and by serology. Vaccinated birds were completely protected against clinical disease and virus excretion was significantly reduced compared to the unvaccinated controls that all died during the experiment. The reproduction ratio, which is the average number of secondary infections caused by an infectious bird, was significantly lower in the vaccinated group (0.82 (95% CI 0.38–1.75)) than in the unvaccinated group (3.2 (95% CI 2.06–4.96)). Results of this study demonstrate the potential of rHVT-ND vaccine in prevention and control of ND outbreaks.

## 1. Introduction

Newcastle disease (ND) is a highly contagious viral disease affecting birds including domestic poultry and is characterized by depression, respiratory signs, diarrhea, nervous signs, and in layers a drop in egg production. Mortality can reach up to 100% in a flock and according to the World Organization for Animal Health (OIE), the disease is among the most important animal diseases globally [1]. ND is endemic or sporadic depending on geographic areas. ND virus (NDV) which belongs to the *Orthoavulavirus* genus within the *Paramyxoviridae* family was recently renamed as *Avian orthoavulavirus* 1 (AOAV-1), but it is commonly known as avian paramyxovirus 1 (APMV-1). All APMV-1 strains belong to a single serotype. ND is caused only by velogenic strains of APMV-1 (vNDV, [2]). Poultry is mostly affected by NDV strains belonging to Class II, which can be further divided into a continuously growing number of genotypes and subgenotypes. A recently published updated NDV strains classification [3] distinguishes XXI genotypes. Epidemiological investigations showed that among the vNDVs genotype VII strains have the widest geographic distribution, while e.g., genotype V strains remain limited to a few geographic locations [4,5].

Vaccination against ND is applied in many parts of the world. The goal of vaccination, however, differs from region to region. In endemically infected regions vaccination is usually aimed at preventing clinical disease, mortality, and production losses. This has been successfully achieved in many instances with several vaccines [6]. In non-endemic regions, vaccination is applied to prevent outbreaks or further spread of the disease. To achieve this, a vaccine should be able to sufficiently reduce virus shedding and transmission. Transmission of an infectious agent can be expressed by the reproduction ratio, *R*, which reflects the average number of infections caused by a single infected bird. The value of *R* should be lower than 1 to stop the spread of an infectious agent. Although several studies have addressed the ability of vaccination to prevent major outbreaks of different diseases in poultry based on their effect on the reproduction ratio of the pathogen (*R* < 1) [7,8,9,10,11] to our knowledge only one study estimating the effect of ND vaccination on *R* has been published [12].

The fusion (F) protein is one of the most important proteins present on the surface of NDV, allowing the virus to attach and penetrate target cells, and is therefore a key virulence factor for the virus as well as a key protective antigen [2,13]. As a consequence, the F gene is the preferred insert to be used in vector vaccine technology. Herpesvirus of turkeys (HVT) has become a widely used vector for poultry vaccines since it is proven to be a very safe and stable virus through its extensive use worldwide to protect chickens against Marek’s disease and it can accommodate foreign genes. One of the available vector HVTND vaccines expressing the F gene of a genotype I NDV (rHVT-ND) (Vectormune^®^ ND) was shown to provide clinical protection against challenges with several genotype II, V or VII NDV isolates in commercial broilers and layers [13,14,15,16]. In addition, after a single vaccination at hatch, commercial layers were protected against clinical signs, mortality and drop in egg production when challenged at 72 weeks of age, demonstrating the long duration of immunity [16]. It also indicates that high level and long lasting protection against vNDV developed in the face of maternally derived antibodies as the breeders had been vaccinated against NDV multiple times, as is common when vaccination is applied in an ND prevention program. Furthermore, it was shown that the immune response elicited by the rHVT-ND vaccine, Vectormune^®^ ND, includes circulating, but also local antibodies as well as cellular immune response [13]. Virus excretion upon challenge in chickens vaccinated with this vector vaccine was significantly reduced when compared to unvaccinated controls [15,16]. However, the effect of the vaccine on NDV transmission is not known, yet this is crucially important information for the implementation of the vaccine in prevention and control of outbreaks, especially in regions where ND is non-endemic. In this study, the effect of a single rHVT-ND vaccination at day-old was assessed for the reduction of genotype VII velogenic NDV transmission in commercial broilers with maternally derived antibodies to NDV.

## 2. Materials and Methods

### 2.1. Vaccine and Challenge Strain

The cryo-preserved cell-associated rHVT-ND vaccine (Vectormune^®^ ND, Ceva-Phylaxia, Budapest, Hungary) expressing the F protein of the avirulent D26/76 genotype I NDV strain (GenBank accession number: M24692) was used in the study. The vaccine was diluted in the corresponding vaccine diluent to obtain one commercial dose in 200 μL.

The challenge strain was a recent Turkish viscerotropic velogenic NDV (vNDV, isolate id.: D3273/1/16TR; GenBank accession number: MT876630), which is a representative of Class II genotype VIIi according to the classification by Diel et al. [17], and genotype VII.2 according to the most recent genetic classification by an international consortium including all NDV referent laboratories [3]. This strain induces 100% mortality between 4–5 days post-challenge in specified pathogen free (SPF) layers challenged at 2–3 weeks of age with a dose of 5.0 log_10_ELD_50_ intra-nasally (Ceva’s unpublished data). The challenge virus stock was diluted in sterile phosphate buffered saline (PBS) to obtain 5.0 log_10_ELD_50_ in a dose of 100 μL.

### 2.2. Chickens and Housing

Embryonated broiler eggs were purchased from a commercial hatchery in Hungary (genetic line: Ross 308) on the 18th day of embryonation and then hatched under experimental conditions to avoid any contamination with vaccine in the hatchery. Maternally derived antibody level (MDA) of chicks against NDV was 5.5 ± 1.7 log_2_ HI titer (mean ± standard deviation) as measured with haemagglutination inhibition test (HI) against NDV LaSota antigen using standard methodology.

During the post-vaccination/pre-challenge period, chickens were housed in BSL-2 animal facilities (vaccinated and unvaccinated groups in the same room with physical separation) and then transferred to two BSL-3 animal rooms for challenge (Prophyl Kft., Bar, Hungary). Contact chickens were transferred 8 h after completion of the challenge infection of the seeders. The chicks were floor-reared on deep litter consisting of single-use straw pellets. Water was provided through nipple drinkers. A standard commercial feed was provided ad libitum. During the post-challenge rearing period, the seeder (directly infected) and the contact chickens belonging to the same group shared the same area with free movement; waterers and feeders were equally accessible for all the chickens in the same group. Rearing density was three chickens per square meter.

The study was conducted in compliance with the provisions of Directive 2010/63/EU, Hungarian Act No. XXVIII/1998, and the Hungarian Government Decree No. 40/2013. (II.14.) and with the permission of the Hungarian competent animal welfare and ethics authority (approval number: BA02/2000-41/2017). Humane endpoint was determined as the phase when a severely diseased chicken was not able to eat or drink any more (euthanized chickens were counted as mortality on the day of euthanasia). Chickens reaching the humane endpoint and all surviving birds at the end of post-challenge observation period were euthanized by intracardiac injection of sodium pentobarbital (5 g/mL).

### 2.3. Study Design

Day-old chicks were randomly allocated to two groups on the day of hatch (Table 1). One of them was vaccinated with one dose of rHVT-ND vaccine in 200 µL subcutaneously by manual injection. The other group was not treated. Blood samples were taken from 20 chicks on the day of hatch to determine the maternally derived antibody level against NDV; these 20 day-old chicks were euthanized.

Forty two days post-vaccination, broilers of both groups were randomly assigned into two subgroups (i.e., seeder and contact), wing-tagged and serum samples (referred to as pre-challenge samples) were collected from each broiler to determine vaccine-take. In the vaccinated group, 20 chickens were allocated both to seeder and contact subgroups (post-challenge rearing in Room #1), whereas in the non-vaccinated group 21 contact chickens were co-mingled with 16 seeders (Room #2). Seeder subgroups were transferred to BSL-3 facility first and challenged intranasally with a dose of 5.0 log_10_ELD_50_ of the vNDV challenge strain (D3273/1/16TR). Eight hours post-challenge contact subgroups were introduced to the corresponding seeder subgroup.

During the post-challenge observation period of 14 days, clinical signs (i.e., prostration, ruffled feathers, nervous signs like paralysis, torticollis) and mortality were recorded daily; oro-nasal swabs (taken from the choanal slit) and cloacal swabs were collected daily from 1–14 days post-challenge (dpch) with the use of invasive sterile EUROTUBO^®^ Collection swabs (code: 300203; Deltalab, Rubi Barcelona, Spain). Blood samples were collected at the termination of the 14-day observation period (14 dpch).

### 2.4. Serology

The HI test was carried out using 4 HAU of LaSota antigen according to the World Organization for Animal Health (OIE) Terrestrial Manual (Chapter 3.3.14, [18]), with the modification that 50 µL of PBS, serum, antigen, or red blood cell suspension was used instead of 25 µL. Pre-challenge serum samples collected at 42 days post-vaccination were subjected to HI test and also measured with a commercial ELISA kit designed for sensitive detection of humoral immune response to rHVT-ND vaccines (NDV F ELISA kit, code: CK122, BioChek, Reeuwijk, The Netherlands). Post-challenge serum samples collected at the end of the trial were tested with a nucleoprotein ELISA (NP ELISA) affording the possibility of selective detection of humoral immune response to challenge when used for chickens vaccinated exclusively with rHVT-ND (ID Screen^®^ Newcastle Disease Nucleoprotein Indirect kit; code: NDVNP; IDvet, Grabels, France) and HI test using LaSota antigen. Both ELISA tests were performed according to the manufacturers’ recommendations. A threshold of more than 2 log_2_ increase of HI titer during the post-challenge observation period was used for the detection of humoral immune response to challenge [12].

### 2.5. Measurement of Challenge Virus Shedding with One-Step RT-Real-Time PCR

NDV containing material was eluted from swab heads in 1 mL PBS by vigorous shaking in Tissue Lyser II (QIAGEN GmbH, Hilden, Germany). Resulted samples were clarified by centrifugation (900× *g* for 3 min at room temperature). Viral RNA was extracted from the samples using Cador pathogen 96 Qiacube HT kit (QIAGEN GmbH, Hilden, Germany) according to the manufacturer’s instructions. Two microliters of RNA extract were used as template for the real-time one-step RT-PCR (TaqMan FastVirus 1-Step Master mix, Applied Biosystems by Thermo Fisher Scientific, Vilnius, Lithuania). Primers and the probe used for the amplification and detection of a fragment of M gene were described previously [19]. Serial dilutions of the challenge strain with a known virus titer were tested to check the sensitivity and the repeatability of the used one-step reverse transcription-real-time polymerase chain reaction (RT-qPCR) in each run and to allow comparison of NDV load based on Ct values. The positivity limit was set at 36 Ct, according to the guidelines of the PCR kit from which the methodology was adopted (TaqMan^®^ NDV reagents and controls, one-step qRT-PCR for NDV RNA, Applied Biosystem, Foster City, CA, USA). Differences in virus shedding were tested between groups with a Mann-Whitney U test on the area-under-the-curve of 36 Ct values.

### 2.6. Quantification of Transmission

Transmission can be quantified by the reproduction ratio (R), which is the average number of new infectious animals produced by one infectious animal during its entire infectious period (*T_inf_*). The reproduction ratio (R) is estimated by separately estimating the transmission coefficient β and the infectious period *T_inf_* and multiplying them:(1)$$R=\beta \;T_{inf}$$

We estimated the parameters using the standard stochastic SIR model [20]. In the SIR model animals are classified into one of three infection states: susceptible (S), infectious (I) or recovered (R). Animals proceed from the S-state after infection to the I-state and by recovery to the R-state. Susceptible animals become infected with a rate *β*
$\frac{I}{N}$ (cases per unit of time), where *I* depicts the number of infectious animals and *N* is the total number of animals (Figure 1).

The transmission coefficient β is estimated by a generalized linear model with a complementary log-log link function [21].
(2)$$\log\left(-\log\left(E\left(\frac{C}{S}\right)\right)\;\right)=\log\;\left(\beta \right)\;+\log\left(\frac{I}{N}\Delta t\right)\;$$

In this equation, *C* is the new cases, which is the number of chickens changing from susceptible state *S* to infectious state *I*.

The infectious period was estimated using parametric survival analysis assuming a normally distributed infectious period, in which the infectious period of each chicken was determined by the number of days that this chicken was classified as infectious.

The confidence interval of the reproduction number was calculated assuming independence of the infectious period and transmission coefficient, which overestimates the width of the confidence intervals [22].
(3)$$var\left(\log\left(R\right)\right)=var\left(\log\left(T_{inf}\right)\right)+var\left(\log\left(\beta \right)\right)$$

Classification of animals to the three states S, I and R was done as follows. All contact birds were assumed to start as susceptible (S) and seeder birds were assumed to be non-susceptible from 1 dpch onwards until becoming infectious. Four birds in the vaccinated seeder bird groups never became infectious and did not show sero-conversion due to challenge. These birds were assumed to be susceptible and treated as contact birds since for them the take of direct infection could not be verified. Recovered birds cannot become infectious or susceptible anymore. Correct evaluation of the level of excretion of virus is required for a consistent classification of I and R states. Chickens with a single weak positive signal in the RT-qPCR based on the Ct-value, were assumed to be not infected and not-infectious. Such signals can indicate environmental contamination or passage of the virus without amplification in the chicken. Therefore, a cutoff Ct-value of 34 was used to classify a chicken as being infected or not. If a seeder or contact chicken had a Ct-value below 34 in the oro-nasal swab sample or cloacal swab sample on two consecutive days, it was considered infectious (I) starting from the first positive day. If the chicken had two or more consecutive days with Ct values above the cutoff in both samples it was considered being recovered (R), again starting from the first day. In cases where Ct values above the cutoff were found for a single sampling day directly before and a single sampling day directly after a positive day (i.e., Ct < 34), the chicken was also considered to be positive on those days, extending the I-state for these birds. We used exactly the same infection state classification criteria for vaccinated and control animals to exclude any influence of the cutoff on the comparison between the vaccinated and unvaccinated group.

The classification was confirmed by the NP ELISA-test at the end of the trial, which selectively detects the humoral immune response to challenge in the vaccinated group, but does not detect the immune response to vaccination. All chickens that had a positive ELISA test had also been tested positive by PCR according to the selected cutoff.

All analyses were performed using R [23] version 3.6.0 with package survival version 3.1.18. (Appendix A).

## 3. Results

### 3.1. Pre-Challenge Serology

Proper vaccine-take was verified by measurement of antibody level in pre-challenge serum samples collected at 42 days post-vaccination (Table 2). The ELISA method used was designed to detect the humoral immune response to the F protein of NDV sensitively; therefore, it has better sensitivity for detection of immune response to rHVT-ND vaccines compared to classical NDV antibody ELISA methods or HI test. All vaccinated broilers showed high ELISA titers (titer range was from 8222–33,593; positivity limit of test is 993 or greater), whereas all non-vaccinated broilers proved to be negative (titer range between 1 and 470). HI test showed 95% positivity in the vaccinated chickens and only low to moderate HI titers (range from 3–7 log_2_) among the sero-positives (positivity limit was above 2 log_2_). All non-vaccinated chickens were seronegative with HI test (range from 0–2 log_2_ HI titer).

Seeder and corresponding contact subgroups had comparable pre-challenge antibody levels (Appendix A).

### 3.2. Clinical Protection

No clinical signs or mortality attributable to velogenic NDV infection was seen in the vaccinated group during the 14-day post-challenge observation period, regardless of the subgroup.

In contrast, all non-vaccinated chickens succumbed to the challenge. Mild clinical signs (prostration and ruffled feathers) appeared at 4 to 6 dpch and mortality followed between 6 and 9 dpch in the seeder subgroup (Figure 2). First clinical signs were recorded between 6 and 9 dpch in the contact subgroup, followed by mortality between 8 and 13 dpch. Altogether three chickens showed nervous signs (torticollis or paralysis); one of them was in the seeder subgroup, the other two in the contact subgroup.

### 3.3. Kinetics of Shedding and Reduction of vNDV Excretion

#### 3.3.1. Kinetics of Shedding in Non-Vaccinated Chickens after Direct or Contact Infection

All unvaccinated challenged chickens showed a strong shedding of challenge virus (Figure 2). Not only the live birds, but the ones found dead on the given day (and also the ones euthanized) were sampled for shedding measurement each day. In case of seeders, oro-nasal shedding was first detected at 1 day post-challenge (3 birds out of 16). The proportion of excreting birds and their viral load increased up to 7 days post-challenge, when mortality reached its peak. Cloacal shedding showed a slight delay compared to oro-nasal shedding as it was detected from 3 days post-challenge. High NDV levels were reached in the oro-nasal swabs more rapidly than in the cloacal swabs. Results of the two swab types became comparable from 6 days post-challenge. There was a delayed onset of shedding in contacts compared to directly infected seeders: oro-nasal shedding was first detected at 4 dpch and cloacal shedding at 6 dpch. Shedding peaked at 9–11 dpch, which coincided with the peak in mortality. The seeder subgroup excreted significantly more in cloacal swabs (Mann-Whitney U = 21, *p* = 0.0002) and in oro-nasal swabs (Mann-Whitney U = 39, *p* = 0.006) compared to the contact subgroup.

#### 3.3.2. Kinetics of Shedding in rHVT-ND Vaccinated Chickens after Direct or Contact Infection

In the vaccinated seeders only a few positive cloacal samples were found throughout the observation period (Figure 3), whereas oro-nasal shedding was detectable in the majority of these chickens. Considering the cloacal shedding there was one seeder out of 20 that had stronger shedding compared to the rest of the group (Figure 3, and Appendix A bird id. 7311). This bird was seronegative in the pre-challenge HI test (1 log_2_ HI titer), although the ELISA showed an immune response to vaccination. Despite this weaker humoral immune response compared to the other group-mates, this chicken was fully protected against clinical signs. Moreover, although oro-nasal and cloacal shedding of this bird was higher than in the other vaccinated seeder birds, it was clearly lower than in the unvaccinated chickens. Four seeder vaccinated birds did not comply with the criteria of being considered an “I” bird for the transmission analysis, although all unvaccinated seeders met this criterion. First shedding birds were detected at 2 dpch, which showed one day delay compared to unvaccinated chickens and oro-nasal shedding peaked at 5–6 dpch. This was followed by a decreasing tendency up to 12 dpch. Lastly there were only one to three weakly positive chickens (oro-nasal swab results of Ct higher than 34) out of 20 chickens at the latest days of follow up (12–14 dpch). Unlike the controls, the shedding birds in vaccinated contact subgroup showed significantly weaker shedding in oro-nasal swabs compared to the shedding birds in the vaccinated seeder subgroup (Mann-Whitney U = 9, *p* = 0.009), whereas no significant difference was found in cloacal samples (Mann-Whitney U = 1, *p* = 0.19), where shedding was hardly detectable in both subgroups. As with the seeders, one contact chicken out of 20 developed weaker immune response to vaccination and stronger shedding (Figure 3, and Appendix A bird id. 7342). This bird was RT-qPCR positive for 8 days and had strong oro-nasal and cloacal shedding compared to the rest of this subgroup, although shedding was still reduced compared to non-vaccinated chickens. All other chickens (95% of contact subgroup) showed markedly suppressed or non-detectable vNDV replication, only 40% of contacts met the criteria of “I” bird.

#### 3.3.3. Effect of Vaccination on vNDV Excretion

The vaccinated group shed significantly less virus than the unvaccinated group both in oro-nasal (Mann-Whitney U = 426, *p* = 1.9 × 10^−6^) and cloacal samples (Mann-Whitney U = 180, *p* = 0.0002). Within the contact subgroups, a significant effect of vaccination was also found both in the oro-nasal (Mann-Whitney U = 82, *p* = 0.002) and cloacal samples (Mann-Whitney U = 62, *p* = 0.002). Excretion was also significantly weaker both in oro-nasal swabs (Mann-Whitney U = 140, *p* = 0.00014), and in cloacal swab samples (Mann-Whitney U = 29, *p* = 0.03) of vaccinated compared to unvaccinated seeders.

### 3.4. Humoral Immune Response to Challenge Infection

Humoral immune response to challenge could be analyzed only in the vaccinated group, since all unvaccinated chickens died during the post-challenge observation period. Selective measurement of antibodies induced by the challenge infection (anti-NP antibodies that are not induced by this type of vaccine) showed 12 positives out of 20 tested seeders and 5 out of 20 contacts. HI test, using LaSota antigen showed the effect of challenge in case of 5 out of 20 seeders and 2 out of 20 contacts (Appendix A). All chickens with strong shedding showed 5–9 log_2_ increase of HI titer, indicating the suitability of LaSota antigen to detect the immune response to challenge even though the challenge virus belongs to genotype VII.

### 3.5. Infectious Period and Transmission Control

#### 3.5.1. Infectious Period

The infectious period was estimated from the contact birds, as they went through a natural infection process and they were thus indicative of infection in the field (Figure 4). Unvaccinated contact birds had a mean infectious period of 4.8 days (95% CI 4.2–5.4), which was significantly longer (*p* = 0.03) than the infectious period of 3.4 days (95% CI 2.2–4.6) for the vaccinated contact birds. More than half of vaccinated contacts neither became infectious (60%) nor sero-converted to challenge (70%; Appendix A). Moreover, the infectious period of unvaccinated birds ended in death, while all the vaccinated birds survived without clinical signs.

#### 3.5.2. Transmission Parameters

The transmission rate parameter β was 0.67 (95% CI 0.42–1.01) infections per infectious bird per day in unvaccinated birds and was significantly (*p* = 0.03) reduced by vaccination to 0.24 (95% CI 0.12–0.47). This indicates that the spread of the infection between vaccinated broilers was more than twice as slow compared to the unvaccinated birds.

The reproduction ratio *R*, i.e., the average number of new infectious animals produced by one infectious animal, for unvaccinated animals was estimated at 3.20 (95% CI 2.06–4.96), which is significantly above the threshold for epidemic spread, while in the vaccinated animals the reproduction ratio was 0.82 (95% CI 0.38–1.75) for which the confidence interval includes the threshold for epidemic spread.

## 4. Discussion

In this study, we examined the effect of a single rHVT-ND vaccine administration at day-old on the transmission of genotype VII vNDV within a group of commercial broilers with maternally derived antibodies. As expected, the vaccinated birds did not show any clinical signs after challenge and virus excretion in the vaccinated group was significantly reduced when compared to the unvaccinated group. Here we showed for the first time for a recombinant HVT-ND vaccine that the reduced virus shedding is accompanied by a significant reduction of the virus transmission among birds. This reduced transmission demonstrates the suitability of the vaccine for outbreak control and prevention of ND in non-endemic regions.

Little information is available on the ability of ND vaccines to reduce transmission. An eye drop live prime-boost vaccination program showed a slight reduction of NDV transmission to contacts although it was not quantified by the authors [24]. Homologous and heterologous experimental inactivated vaccines were tested for shedding reduction [25]. Van Boven et al. [12] showed that vaccination using two live attenuated vaccines (Ulster, or LaSota) in a prime-boost regime sufficiently reduced transmission (*R* < 1) when a HI titer higher than 3 log_2_ was reached in at least 85% of the vaccinated population. However, this level of protection is hardly ever achieved in broiler chickens under field conditions, possibly due to poor vaccination practices, inadequate vaccination schedules, or maternal antibodies interference, among others [26,27]. Consequently, there is a need for vaccines that could both effectively reduce transmission when applied in chickens with MDA and be easily and consistently mass-administered under field conditions. The vaccine technology using Marek’s disease vaccine strain HVT as a vector expressing the F protein of NDV was shown to successfully immunize day-old chickens or 18-day embryos in the presence of maternally derived antibodies [28]. In addition, hatchery vaccination was shown to elicit the expected immune response more efficiently and more consistently compared to on farm vaccinations in a highly pathogenic avian influenza control framework, which is a key when dealing with fatal and contagious diseases [29]. Kinetics of immune response to rHVT-ND shows a gradual building up of immunity. Strong clinical protection with significant shedding control can be achieved at 3 weeks of age in commercial broilers with MDA, which is followed by a total clinical protection, and a strengthening suppression of challenge virus replication [15]. The recent study aimed the setting of a model for transmission control analysis and determination of the *R* value that can be reached by the rHVT-ND vaccine, therefore the age at challenge was set at the strongest expected protection level in broilers with MDA.

The rHVT-ND vaccine was able to markedly reduce the transmission of NDV infection within the vaccinated group and the point estimate of *R* was below 1, indicating that the infection will fade out after introduction in a vaccinated population. As such the vaccine is suitable for use in vaccination programs to eliminate NDV from chicken populations and prevent outbreaks upon introduction of the virus. However, the confidence interval of *R* includes 1, so we cannot exclude that *R* in vaccinated birds is above 1. Given the value of 0.82, an experiment with sufficient power to establish whether the true value of *R* is significantly below 1 would include a huge number of animals, which would make it unfeasible to perform and be ethically questionable because of the large number of severely suffering control animals. In addition, we think it is not needed, because the results from the virus excretion suggest that *R* in the vaccinated group could have been overestimated. Virus excretion in the vaccinated seeder birds was significantly higher than in the vaccinated contact birds, indicating that the contact birds were exposed to a higher virus load than what would be normal in a flock in the field where birds are only exposed to virus from contact infected birds. An extended transmission experiment, where a second generation of contact birds is included, would be needed to confirm this [30]. As a next step it would be more appropriate to examine the effect of the vaccine on transmission under field circumstances. The reason is that the effectiveness may be reduced by farm’s specific factors, such as the age at infection, ratio of infected seeder birds at the beginning of infection chain, co-infections, indoor environment conditions, and bird density. However, even if *R* in vaccinated birds would be slightly above 1, the effect on ND outbreaks in the field would be considerable. For example, the probability that a virus introduction into a flock will result in an outbreak equals (1–1/*R*), meaning that most virus introductions would not result in an outbreak in vaccinated broilers as opposed to 69% infected chickens (95% CI 51−80%) in unvaccinated flocks with an *R* of 3.20 (95% CI 2.06–4.96) [20]. Moreover, the proportion of the flock affected by such an outbreak would be considerably smaller. In a vaccinated flock, a few birds would become infected during an outbreak against an expected 96% in an unvaccinated flock [20]. In addition, the infection spread is slower which allows for more time to intervene, because infection rate per day was reduced by more than two fold, moving from 0.67–0.24. Finally, the combined effect of fewer outbreaks and smaller outbreaks would likely reduce between flock and between farm transmission, possibly to an effect that transmission would be stopped. This is exemplified by Aujeszky’s disease which has been eradicated from the pig population in several countries in Europe by vaccination, even though the vaccination scheme in finishing pigs (the pig group comparable with broilers) could only reduce transmission to *R* = 1.5 [31]. Our results could be used to parameterize a mathematical transmission model to examine the effect of the vaccine in a regional or national vaccination program, while simultaneously examining the effect of vaccination coverage on between farm transmission [32].

Vector rHVT-ND vaccines have shown the ability to induce a strong clinical protection against ND along with significant shedding reduction. This high level of clinical protection against genotypes being heterologous to the vaccine insert was observed for NDV challenge strains representing genotype II, genotype V or genotype VII [13,14,15,16]. The assessment of shedding was based on measurements at a few selective days focusing on the acute phase of the disease after direct challenge of chickens in these publications. To the best of the authors’ knowledge, there is no published study (i) describing the kinetics of vNDV shedding in rHVT-ND vaccinated and unvaccinated chickens on a daily basis, (ii) comparing the cumulative amount of challenge virus shed by vaccinated chickens that survive the challenge with the unvaccinated controls that usually die within seven days post-challenge and (iii) quantifying the effect of vaccination on transmission.

The genotype of the challenge virus (genotype VII) is heterologous to the donor of vaccine insert (genotype I) and is the most widely prevalent group of NDV threatening the poultry industry. In the regions where velogenic NDV is present, the vaccination program of commercial broilers traditionally includes several applications of live or of live and killed ND vaccines to provide early and strong protection. In this study, the potential of a rHVT-ND vaccine alone was assessed at an age where maximum level of protection regarding the transmission control is expected and the MDA is completely decayed from the unvaccinated hatch mates. RT-real-time PCR was used for quantification of vNDV replication and shedding. Since we do not have exact information on the correlation of live virus titer and NDV RNA load in the swab samples throughout the time course of shedding, no extrapolation to EID_50_ was used in the analysis, but comparison of vNDV amount among the samples and also the categorization of chickens for the SIR transmission model was done based on Ct values.

The course of infection in the control group indicates that the experimental design is a good representation of the infection in a flock. All the unvaccinated seeders and the unvaccinated contacts showed strong shedding, contacts shed even significantly less challenge virus than the seeders showing that the direct challenge was strong enough to mimic the field infection coming from unvaccinated flocks. Moreover, the large proportion (80%) of vaccinated seeder birds that became infected demonstrates a good natural challenge of the vaccinated contact birds. It would have been interesting to know whether most of the contact infections in the vaccinated group were caused by the single bird with the relatively high level of excretion or weak shedders also contributed significantly. In addition, if the post-challenge period had been somewhat longer, more birds might have stopped shedding and seroconverted, adding to the information on the infection of individual birds. However, here our goal was to examine the protection induced by vaccination at day-old against transmission at the end of the broiler production period thereby limiting the possible length of the experiment, with associated welfare problems due to high body weight of the broilers. The resistance of vaccinated chickens was reflected by the delayed onset of shedding, significantly reduced amount of virus shed, and the shorter infectious period of vaccinated contacts (Figure 4). In the case of vaccinated chickens, cloacal shedding was more reduced compared to oro-nasal shedding, possibly due to the strong cellular and humoral immunity that efficiently limits challenge virus dissemination in the body. This finding is in line with previous publications [13,16]. The NP ELISA used to follow the humoral immune response to challenge proved to be a better indicator of challenge virus replication compared to the HI test using LaSota antigen. This shows the value of NP ELISA as a DIVA test for field virus infection monitoring in those flock that received only rHVT-ND vaccine to protect against ND.

The assessment of ND vaccine efficacy and coverage is usually monitored by HI test as requested by veterinary authorities. This requirement has been set for conventional live vaccines, which mode of action is completely different from rHVT-ND vaccines. The latter ones generally only express the fusion protein of NDV, while HI test is adequate to measure the anti-haemagglutinin-neuraminidase (HN) antibodies. Due to steric hindrance anti-F antibodies are able to hide the NDV HN protein and through this are able to inhibit the agglutination observed in an HI test. Even though low HI titers can be measured, this method is not sensitive enough to monitor the efficacy of rHVT-ND vaccination, as supported by previous publications [15,16]. Conventional ELISA kits for anti-NDV antibody detection are also weakly sensitive for monitoring rHVT-ND elicited immunity [15,16]. New ELISA kits that were optimized for the detection of humoral immune response to this type of ND vaccines through the increased sensitivity in anti-F antibody detection [11,33] provide high ELISA titers in rHVT-ND vaccinated flocks and are capable of detecting the humoral immune response earlier than the HI test. In this study, all vaccinated chickens had high F ELISA antibody titers while part of them showed a HI titer of 3 log_2_ or below. All of them were fully protected against clinical signs of ND and significantly suppressed challenge virus replication. This provides evidence that low HI titers should not be taken as indication of lack of immunity, and that monitoring tools should be adjusted to the type of vaccine.

## 5. Conclusions

This experimental study shows the potential of rHVT-ND to prevent and control ND outbreaks even when they are caused by a virus belonging to a heterologous genotype than the vaccine insert. Further research is needed to establish the effectiveness of the vaccine at earlier infection ages that is when the field virus is introduced in a population that is not fully immunized. It should also be highlighted that rHVT-ND is not normally applied as a single ND vaccine in areas with high ND virus pressure. The combination of rHVT-ND with live vaccines that is the usual prevention strategy in endemic areas, will probably more efficiently reduce the transmission rate.

In conclusion, the results of this study clearly demonstrate the potential of rHVT-ND to prevent and control ND outbreaks in broilers.

## Figures and Tables

**Figure 1 vaccines-08-00614-f001:**
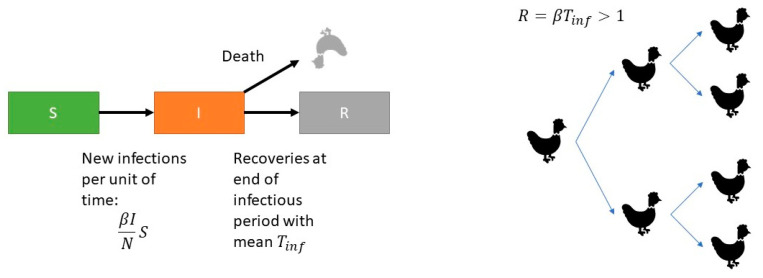
Diagram of the transmission model. Infection occurs with a rate proportional to the fraction of infectious birds *β*
$\frac{I}{N}$ and birds recover or die at the end of infectious period with mean *T_inf_*. Multiplication of the transmission coefficient and the mean infectious period *T_inf_*. is the basic reproduction number *R*. For *R* > 1 an exponential growing infection will occur in a susceptible population and for *R* < 1 the infection will go extinct.

**Figure 2 vaccines-08-00614-f002:**
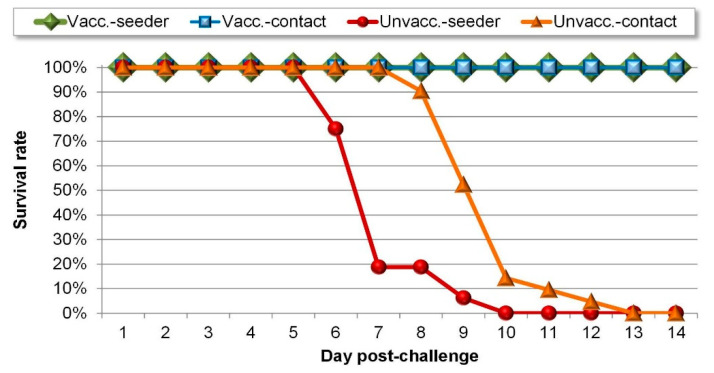
Time course of mortality in the different subgroups.

**Figure 3 vaccines-08-00614-f003:**
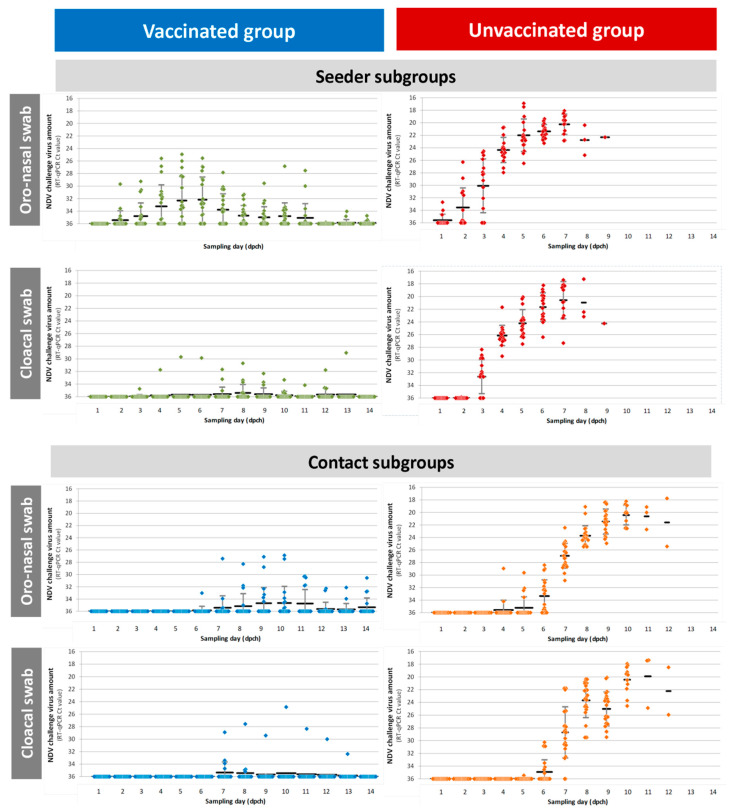
Kinetics of velogenic Newcastle disease virus (NDV) shedding in vaccinated and unvaccinated birds after direct infection (seeder subgroups) or contact infection (contact subgroups). Seeder subgroups were challenged with 5.0 log_10_ELD_50_ of velogenic NDV intra-nasally. Contacts were co-mingled with seeders from 8 h post-challenge. Oro-nasal swabs and cloacal swabs were collected daily for 14 days post-challenge; vNDV amount was quantified by RT-qPCR. Individual results, and mean and STD of Ct values are shown.

**Figure 4 vaccines-08-00614-f004:**
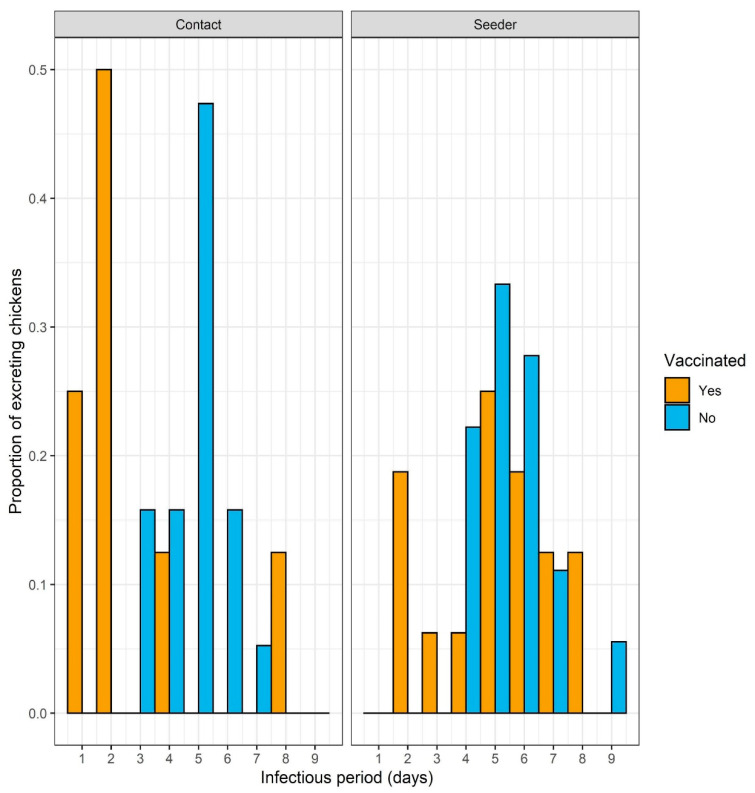
Distribution of infectious periods of the contact (left panel) and seeder (right panel) birds. In the vaccinated contact groups the infectious period of two birds were censored from the analysis at day 14 dpi. Both were infectious for one day.

**Table 1 vaccines-08-00614-t001:** Study design to quantify the effect of a turkey herepesvirus vector-based vaccine on the transmission of Newcastle disease virus in broilers.

Group	Subgroup	Vaccination	Challenge	Samplings
Vaccinated	Seeder (Vacc.-seeder)	rHVT-ND vaccine	Intra-nasal (*n* = 20)	0 dpch (*n* = all): Blood sampling 1–14 dpch daily (*n* = all): Oro-nasal and cloacal swabs Clinical observations 14 dpch (*n* = all): Blood sampling
	Contact (Vacc.-contact)		Contact (*n* = 20)	
Unvaccinated control	Seeder (Unvacc.-seeder)	No	Intra-nasal (*n* = 16)	
	Contact (Unvacc.-contact)		Contact (*n* = 21)	

Note: Commercial broilers with MDA were randomly assigned into two groups at day-old and were vaccinated with a turkey herpesvirus vectored ND vaccine (rHVT-ND), or remained unvaccinated. Challenge infection was performed at 42 days post-vaccination with velogenic Newcastle disease virus Genotype VII.2 (formerly genotype VIIi); D3273/1/16/TR strain; a dose of 5.0 log_10_ELD_50_/bird was intra-nasally administered to seeder subgroups. Unchallenged contact group-mates were co-mingled with seeders from 8 h post-challenge onwards. Post-challenge samplings and clinical observations were terminated at 14 days post-challenge (dpch).

**Table 2 vaccines-08-00614-t002:** Pre-challenge serological results and clinical protection against vNDV challenge.

Group	Pre-Challenge Serology		Clinical Protection **
	ELISA * Titer (mean ± SD Range and Positivity)	Log_2_ HI * Titer (mean ± SD Range and Positivity)	
Vaccinated	25,114 ± 5254 (8222; 33,593) 100%	3.9 ± 1.2 (1; 7) 95%	100%
Unvaccinated control	146 ± 111 (1; 470) 0%	0.9 ± 0.7 (0; 2) 0%	0%

Note: * BioChek ND F ELISA and haemagglutination inhibition test (HI) were used to verify vaccine-take. ** Clinical protection is the percentage of chickens that survived the 14-day clinical observation period without showing clinical signs indicative of velogenic NDV (vNDV) infection.

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
