# Peer review of "A Herpesvirus of Turkey-Based Vector Vaccine Reduces Transmission of Newcastle Disease Virus in Commercial Broiler Chickens with Maternally Derived Antibodies"

_vaccines, 2020, doi:10.3390/vaccines8040614_

Round 1

Reviewer 1 Report

The manuscript entitled "A Herpesvirus of turkeys-based vector vaccine reduces transmission of Newcastle disease virus in commercial broiler chickens with maternally derived antibodies" by Tatár-Kis et al., described the impact of vaccine (rHVT-ND) on NDV transmission using the reproduction ratio (?). the manuscript well written with thoughtful discussion and interesting ideas. However,i can see the main value of the used mathematical model is to assess and assist in setting up a control plan for outbreaks in non-endemic regions rather than evaluating the vaccine efficiency that has its stander evaluation procedures and the currently used study not cover all aspect. Such transmission can be affected by many other factors one of them is the vaccine, wondering if more factors could be included in the currently used model. This should be highlighted in the discussion section.

-The Table S2 file is not working for me. 

Line 153: did you use a serial dilution of the challenge strain with known viable virus titer to indicate the sensitivity and repeatability of used real-time PCR system in each PCR run?  

Line 166:a schematic presentation of the transmission model used should be included to make section 2.6. clear.

Line 366: It would be great if there is a group where vaccinated noninfected birds introduced to the infected nonvaccinated group, where you can measure transmission from infected non vaccinated (common in the field with a high level of virus excretion) to vaccinated noninfected. is there is previous data in this regard that you can use to support the discussion?

Line 380-391: from previous studies, correlate between the higher and homologous antibody titer (humoral immunity) and virus shedding and clinical protection. comparing to adequate cellular and acceptable humoral immunity and virus shedding and clinical protection should be highlighted in the discussion section.

Line 427: please mention why the conventional ELISA kits for anti-NDV antibody detection is not sensitive for monitoring rHVT-ND elicited 428 immunity?

Reviewer 2 Report

Major comments

The present study evaluated the transmission rate in vaccinated and non-vaccinated commercial broiler chickens. The vaccination was performed with a single dose of a recombinant herpesvirus of turkey vaccine expressing the fusion protein of Newcastle disease virus (NDV) in day-old chickens with maternally derived antibodies (MDA). Vaccinated birds had antibody titers measured by ELISA and HI tests before the challenge. Authors showed that the vaccine could induce clinical protection, reduction in virus shedding, and transmission to contact birds after challenge forty-two days post-vaccination. They aimed to evaluate when the maximum level of protection in terms of transmission is expected, and the MDA is completely decayed in unvaccinated birds. They then evaluated the effectiveness of vaccination at day-old commercial chickens against transmission at the end of the broiler production period. However, meat broilers are usually slaughtered at 42 to 47 days of age in the biggest poultry meat producer countries (USA, China, Brazil, the EU). The challenge performed in this study was done in 42 days-old chicken broilers, and the mean death time of the directly infected and contact birds was 7 and 9.7 days, respectively. It is not clear why they assessed the protection and transmission rate at the age of the slaughter. The main challenge for the meat broiler industry is to obtain early protection with the decay of MDA. A few studies showed the age of challenge when using a recombinant HVT vaccine is also very important. The clinical outcomes from birds vaccinated with recombinant vaccines vary depending on the age of the challenge. The decay of MDA and the incomplete immune response induced by a recombinant vaccine before the slaughter is a great challenge for the broiler industry. Other studies also compared live vaccines with recombinant HVT to overcome this incomplete immune response from recombinant HVT vaccines in a challenge at 21-28 days of age. The authors should discuss that providing references.  

Additionally, the HI test using the challenge strain and NP-ELISA should be performed to evaluate if the induced immunity could reduce the challenge virus replication comparing the antibody levels before and after the challenge. The authors seemed to provide information about the NP-ELISA after the challenge in Table S2. However, this reviewer could not open the provided table. 

The present study is relevant to the field as they quantified the transmission rate and the efficacy against an NDV challenge induced by a recombinant HVT vaccine in day-old commercial birds with MDA. 

Other comments

Lines 89-90, page 2: Provide the reference that the D3273/1/16TR induces 100% of mortality in SPF birds at 2-3 weeks of age. 

Lines 93 to 96: Describe how the MDA was accessed, for example: from 20 chicks by HI test using LaSota strain. 

Line 117-118: Clarify if those chicks were used in the experiment or they were euthanized

Line 125: Why did you add the contact birds only eight hours after the challenge?

Line 144-145: It is not expected to have high HI titers after a vaccination using a recombinant HVT-F, but relatively high titers (log6 and 7) in some birds. As the authors discussed, the F antibodies can only inhibit the agglutination because they high the HN protein. That is why it is essential to describe very well the used methodology. Please provide more details about the HI methodology adding the reference.

Page 145: What was the antigen used before the challenge? It will be interesting to get the HI titers before and after the challenge in surviving birds with the challenge strain. It will be essential to check if there will be an increase or decrease of HI titers after challenge.

Where are the results of the NP-ELISA after the challenge? 

Lines 193-196. The four birds should be excluded from the experiment, as the challenge strain intake was not confirmed. I agree that they could be assumed to be infected, but something could also happen during the infection. Were these four birds placed in the same isolator?

Table 2: ELISA titer's positivity seemed to be 0%, not 95% in the unvaccinated group before the challenge. Please check this information.

Lines 238-243: It will be interesting to calculate the mean death time (MDT) in both contact and directly infected non-vaccinated birds. 

Lines 249 to 261: Shedding by cloacal route was higher than the oral route in contact birds none vaccinated at 5 to 7 dpch. Can you explain this?

Page 225-225. This study also evaluated the transmission. 

Lines 286-289: Many studies compared the challenge virus shedding in vaccinated and non-vaccinated birds. 

Line 391: This study did not evaluate the virus dose need for infection. Please remove this sentence.

Lines 414-417: This should be moved to the results section. 

Lines 433-434: The HI titers before and after the challenge using the challenge strains are the best indicators to evaluate a decrease in challenge virus replication. Unfortunately, this study did not assess that. 

Lines 440-441: Authors conclude that "combination of rHVT-NDV with live vaccines will be more efficient to reduce the transmission rate". However, this study did not evaluate the combination of recombinant rHVT-NDV with live vaccines. 

Table S2 had an invalid format file. 

Round 2

Reviewer 2 Report

The authors improved the current version of the manuscript after the reviewers' comments. The supplementary table 2 was now available. Therefore, this reviewer does not have any additional comment.